# Effect of Snowmelt on Groundwater Bacterial Community Composition and Potential Role of Surface Environments as Microbial Seed Bank in Two Distinct Aquifers from the Region of Quebec, Canada

**DOI:** 10.3390/microorganisms11061526

**Published:** 2023-06-08

**Authors:** Karine Villeneuve, Valérie Turcotte-Blais, Cassandre Sara Lazar

**Affiliations:** Department of Biological Sciences, University of Québec at Montréal, UQAM, C.P. 8888, Succ. Centre-Ville, Montreal, QC H3C 3P8, Canada; villeneuve.karine.5@courrier.uqam.ca (K.V.);

**Keywords:** aquifer, groundwater recharge, groundwater microbial ecology, snowmelt, bacterial communities, perturbation

## Abstract

Events of groundwater recharge are associated with changes in the composition of aquifer microbial communities but also abiotic conditions. Modification in the structure of the community can be the result of different environmental condition favoring or hindering certain taxa, or due to the introduction of surface-derived taxa. Yet, in both cases, the local hydrogeochemical settings of the aquifer is likely to affect the amount of variation observed. Therefore, in our study, we used 16S rRNA gene sequencing to assess how microbial communities change in response to snowmelt and the potential connectivity between subsurface and surface microbiomes in two distinct aquifers located in the region of Vaudreuil–Soulanges (Québec, Canada). At both sites, we observed an increase in groundwater level and decrease in temperature following the onset of snow melt in March 2019. Bacterial community composition of each aquifer was significantly different (*p* < 0.05) between samples collected prior and after groundwater recharge. Furthermore, microbial source tracking results suggested a low contribution of surface environments to the groundwater microbiome except for in the months associated with recharge (March 2019 and April 2019). Overall, despite differences in soil permeability between both sites, the period of snow melt was followed by important changes in the composition of microbial communities from aquifers.

## 1. Introduction

Microorganisms are ubiquitous in aquifer systems, where they drive organic and inorganic compound transformations and thus control biogeochemical cycles [1]. Therefore, the quality of groundwaters and the ecosystem services they provide are greatly influenced by the composition of the microbial communities they harbor [2]. Owing to protective overlaying layers of soil and greater water residence time, it was previously assumed that these communities and the abiotic conditions of aquifers were relatively stable over time [3]. Yet, several recent studies revealed important seasonal dynamics of environmental conditions and aquifer microbiomes composition, notably as a result of groundwater recharge events [4,5,6,7]. Aquifers represent the largest reservoir of liquid freshwater on Earth. About 2.5 billion people worldwide depend solely on groundwater as a source of freshwater [8,9]. In the province of Quebec, Canada, approximately 20% of the population relies on groundwater for its drinking water [10]. As this number is expected to increase in the future, understanding and predicting the responses of microbial communities to seasonal variation is critical to ensure safe access to this resource.

Aquifer systems can be viewed as open biogeoreactors connected to terrestrial and aquatics ecosystems through processes of groundwater recharge and discharge [2,5]. While the former process refers to water flowing into the aquifer, the latter refers to groundwater exiting the subsurface. Regarding aquifer microbiomes, the process of groundwater recharge was previously associated with variations in biotic and abiotic conditions of aquifers, and thus considered a disturbance [4,5,6,7]. Indeed, following a period of recharge, important changes in the composition of the microbial communities were generally observed. While some attributed most of these differences to the fluctuation of environmental conditions favoring or hindering certain taxa [4,6,11], the introduction of surface-derived taxa was also suggested as a factor responsible for the observed dissimilarity in microbial community composition [5,11,12]. However, it is likely that local hydrogeochemical characteristics of aquifers influence both the variation in abiotic conditions and migration of surface-derived taxa.

The movement of liquids, gases, and nutrients through the soil is notably influenced by its porosity, pore-size, and permeability [13,14]. These characteristics are also likely to influence the presence, taxonomy, and function of microorganism residing in individual pore-spaces [15,16]. Thus, the effect of groundwater recharge on microbial community composition and environmental conditions can be expected to differ between aquifers with different hydrogeochemical settings [17]. Yet, to date, few spatio-temporal studies have assessed and compared the temporal variations associated with recharge in distinct aquifers.

Therefore, in this study, we used 16S rRNA amplicon sequence variants (ASVs) to infer the potential connectivity between surface and subsurface bacterial communities from two aquifers with different hydrogeochemical settings located in the Vaudreuil–Soulanges region (Québec, Canada). The aims of this study were to (1) evaluate the temporal variations in microbial community composition according to seasons and (2) investigate and compare the migration of microorganisms from surface to subsurface and back to the surface.

## 2. Materials and Methods

### 2.1. Study Sites

Two unconfined aquifers located in the Regional County Municipality of Vaudreuil–Soulanges were sampled once a month from September 2019 to August 2020 (Figure 1). The Rigaud site (R; 45°28′49.0″ N, 74°15′37.5″ W) is located downstream of the Raquette river, which flows into the Ottawa river. Contributions of groundwater to river flow were previously established, and this discharge of groundwater into the surface aquatic environment is considered crucial during the dryer summer months [18]. The soil from this site is mainly composed of low-permeability clay which favors surface water runoff rather than infiltration; thus, the aquifer is poorly drained and water recharge is relatively low [19]. The Rigaud site is also influenced by human activities, notably an agricultural field and a large road which borders the sampling location. The Saint-Lazare site (SL; 45°23′05.1″ N, 74°11′49.7″ W) is located in a natural park mainly composed of pine grove. The soil is mostly composed of permeable sand and the aquifer is considered very productive [18]. The site is notably located in the main area of groundwater recharge. Furthermore, a river is located downhill of the aquifer. For both sites, the period of snow cover usually extends from November to April.

### 2.2. Sampling

Groundwater, river, and snow samples were collected in sterilized polypropylene bottles (Nalgene, Rochester, NY, USA). Groundwater samples were collected using a submersible pump (12 V/24 V Mini-Monsoon, Waterra, Mississauga, ON, Canada) according to standard operating procedure [20], discarding a minimum of three times the equivalent of the well volume. During each sampling campaign, 4 L was collected for microbiological analyses. Dissolved oxygen was measured in the field using a YSI multi-parameter probe (model 10102030, Yellow Springs, OH, USA). At both sites, 1 L from the adjacent river was recovered from the shore. Soil and snow were sampled as representatives of aquifer recharge. Soil was collected using a sterilized shovel and stored in 50 mL Falcon tubes. Soil could not be collected from January 2020 until March 2020 due to important layers of snow. Approximately 1 L of snow was collected at both sites in November 2019, January 2020, February 2020, and March 2020. No snow was collected in December 2019 due to warmer temperature which caused the snow to melt. Overall, we collected a total of 71 samples (24 groundwater samples, 21 river samples, 18 soil samples, and 8 snow samples).

All samples were kept in the dark and on ice during transport. Water samples were stored at 4 °C until filtration in the lab, which was performed on the same day as sampling. Snow samples were thawed at room temperature in the lab before filtration, which was also performed on the same day as sampling. In order to separate larger microorganisms from smaller ones, serial filtration was performed (0.2 μm, 0.1 μm) using polyethersulfone filters (Sartorius, Göttingen, Germany). For both filter types, the total amount of collected water was passed through one single filter and subsequently stored at −20 °C.

### 2.3. Physicochemical Analyses

Groundwater temperature and level were collected from probes installed semi-permanently in each well. Level corresponds to the height in meter (m) of the water above the probe.

For measurement of ammonium and ammonia (NH_x_), water samples were collected in plastic scintillation vials after filtration on a 0.2 μm polyether sulfone filter (Sarstedt^®^, Numbrecht, Germany). Samples were analyzed on a Flow Solution 3100 autosampler using a chloramine reaction with salicylate to form indophenol blue dye (EPA Method 350.1). For measurement of nitrates (NO_3_) and nitrites (NO_2_), water samples were collected in plastic scintillation vials after filtration on a 0.45 μm polyethersulfone filter (Sarstedt^®^, Numbrecht, Germany). Samples were analyzed with a continuous flow analyzer (OI Analytical Flow Solution 3100©, OI Analytical, College Station, TX, USA) using an alkaline persulfate digestion method, coupled with a cadmium reactor, following a standard protocol [21]. Finally, to measure dissolved organic and inorganic carbon (DIC/DOC), water samples were collected in gas-free glass bottles after filtration on a 0.45 μm polyethersulfone filter (Sarstedt^®^, USA). Samples were analyzed with an OI Analytical Aurora 1030 W TOC Analyzer (OI Analytical, College Station, TX, USA) using a persulfate oxidation method. All geochemical analyses were conducted at the GRIL-Université du Québec à Montréal, (UQAM, Montréal, QC, Canada) analytical laboratory.

### 2.4. DNA Extraction, Illumina Sequencing, and Sequence Analysis

For all water samples, DNA collected on the polyethersulfone filters was extracted using the DNeasy power water kit (Qiagen, Hilden, Germany). For soil samples, DNA was extracted from 250 mg of soil using the DNeasy power soil kit (Qiagen, Hilden, Germany). Both types of extraction were carried out according to the manufacturer’s instructions. Extracted DNA samples were stored at −20 °C until further use.

Archaeal 16S rRNA genes were not amplifiable for most of the analyzed samples; therefore, we did not sequence archaeal 16S rRNA genes. Bacterial 16S rRNA genes were amplified using the polymerase UCP hiFidelity PCR kit (Qiagen, Hilden, Germany) for better sensitivity of low DNA concentrations. The V3-V4 region of the bacterial 16S rRNA gene was targeted using the primer pair B341F (5′-CCTACGGGAGGCAGCAG-3′)—B785R (5′GACTACCGGGGTATCTAATCC-3′) primer pair [22]. PCR amplification was carried out under the following conditions: denaturation at 98 °C for 30 s, annealing for 30 s at 57 °C, extension at 72 °C for 1 min, and final extension at 72 °C for 10 min. This cycle was repeated 33 times. Sequencing was performed at the CERMO-FC genomic platform (Center for Excellence in Research on Orphan Disease—Foundation Courtois) at UQAM using an Illumina MiSeq 2300 and the MiSeq reagent kit v.3 (600 cycles, Illumina, San Diego, CA, USA). Negative control for the PCR amplifications was also sequenced.

### 2.5. 16S rRNA Amplicon Sequencing and Data Processing

Due to low biomass on the 0.1 μm filter, only a few samples had a sufficient amount of reads to be processed separately. Therefore, fastq files from the 0.1 μm and 0.2 μm filters were combined prior to processing.

Raw sequences were processes in R (v4.2.2) [23] using DADA2 (v1.24.0) for quality filtering, merging paired reads, inference of amplicon sequence variant (ASV), and removal of chimeric sequences, with slight modification of the Callahan et al. [24] workflow. After removal of primers, truncation length was set to 280 bp and 240 bp for forward and reverse reads, respectively, and the maximum number of expected errors allowed in a read was set to 4. ASVs were inferred using pseudo-pooling, which increases sensitivity to rare variants. The R package decontam (v1.16.0) [25] was used to identify and remove contaminant DNA sequences with a prevalence threshold of 0.6. Taxonomic assignment was carried out by mapping sequences against the trained SILVA SSU database (release 138.1) [26]. ASVs that were not classified as bacteria were discarded. Our initial dataset consisted of 5,229,058 reads and following processing our final dataset consisted of 2,482,852 reads. Phylogenetic relationships were inferred using the DECIPHER package (v2.24.0) [27] to align ASV sequences. Fastree (v2.1.11) [28] was used to build a midpoint-rooted phylogenetic tree from the alignment. Sequences were deposited on the National Center for Biotechnology Information platform (NCBI) under the BioProject ID PRJNA912985.

### 2.6. Statistical Analyses

Analyses were computed in R and a significance level of α = 0.05 was used for all statistical tests.

ASVs with a relative abundance of less than 0.005% were considered as sequencing error [29] and were subsequently removed from the dataset. Furthermore, abundances were rarified to an event depth of 2530 which corresponded to the lowest number observed in a sample. To validate the choice of rarefaction, linear regression was used to compare the ASV richness and Shannon diversity between the rarefied and extrapolated datasets (Appendix A Appendix A).

A previous study revealed that bacterial community composition from both sites was dissimilar [30], and therefore all analyses were carried out separately for Saint-Lazare and Rigaud. Dissimilarity in the structure of the microbial communities as a function of habitats and sampling months were visualized using Principal Coordinates Analysis (PCoA) generated from a Bray–Curtis dissimilarity matrix based on the rarefied and Hellinger-transformed relative abundance. To determine if observed clusters were significantly different, we used permutational analysis of variance (PERMANOVA) with default 999 permutation. PCoA and PERMANOVA were processed with the vegan package (v2.6-4) [31] using the vegdist and adonis2 function, respectively. To assess the changes in diversity, Shannon diversity index were calculated for each groundwater samples using the estimate_richness functions of the vegan package and the results were visualized using ggplot2 (v3.4.0) [32].

To evaluate the potential connectivity between surface and subsurface bacterial communities, we used the fast expectation–maximization for microbial source tracking (FEAST) method with R package FEAST (v0.1.0) [33]. FEAST estimates the contribution of input source community to sink community and reports the potential contribution of unknown sources. At both sites, soil, snow, and groundwater samples collected up to three months before a groundwater sample identified as sink were considered as potential sources of diversity. At the Rigaud site only, river samples were also considered as potential sources given the observed groundwater–river connections.

## 3. Results

### 3.1. Fluctuation in Groundwater Level, Temperature, and Physicochemical Qualities

For both aquifers, we observed an important change in groundwater level and temperature during our sampling period (September 2019 to August 2020), which can be attributed to groundwater recharge from snowmelt during the month of March 2020 (Figure 2a–d).

During this event, in the Saint-Lazare aquifer, groundwater levels increased from their lowest daily height of 2.74 m in March to their highest value of 3.78 m in May, while temperature simultaneously dropped from its March highest daily average of 7.31 °C to its lowest recorded temperature of 5.22 °C in April (Appendix A Appendix A). The highest temperature of 7.41 °C was measured in January. Compared to Saint-Lazare, the groundwater level and temperature of the Rigaud aquifer varied more across the sampling period. Nonetheless, we also observed an important increase in groundwater levels from their lowest daily average of 1.12 m in March to their highest recorded value of 2.56 m in April. During the same period, groundwater temperature decreased from its March highest daily average of 6.69 °C to its lowest recorded temperature of 3.61 °C in April (Appendix A Appendix A). The highest temperature of 10.87 °C for this aquifer was measured in November. Following the snow melt, groundwater level and temperature of both aquifers gradually returned to their values prior to snowmelt.

We used this data to distinguish two different periods: pre-recharge, which began from the beginning of our sampling in September 2019 and lasted until February 2020, and post-recharge, which started in March 2020 and lasted until the end of our sampling in August 2020.

Other physicochemical qualities of groundwater from both aquifers were relatively stable across the sampling period, with the exception of a noticeable increase in dissolved inorganic carbon during the month of March for both sites (Appendix A Appendix A).

### 3.2. Temporal Bacterial Community Composition and Diversity

At both sites, visual inspection of the PCoA plot suggested that soil bacterial communities were distinct from other sampled environments (Figure 3a,b). At the Saint-Lazare site, microbial communities from the river and snow clustered together, suggesting more similar communities, while groundwater samples were distantly distributed (Figure 3a). PERMANOVA testing of community composition at the Saint-Lazare site revealed that habitat type significantly explains 51% of the variance (*p* < 0.01; Appendix A Appendix A). At the Rigaud site, river and snow communities formed three loose clusters with groundwater samples from the months of March, April, June, and August (Figure 3b). PERMANOVA testing of community composition for this site indicated that habitat type significantly explains 43% of the variance (*p* < 0.01; Appendix A Appendix A).

To further investigate the temporal variations in bacterial communities in aquifers we generated a second PCoA with only groundwater samples. Visual inspect of this second ordination suggests that the composition of the bacterial community from the pre-recharge period (September, October, November, December, January, and February) is less variable across the different months when compared to the composition of community from samples collected after the recharge event (March, April, May, June, July, and August). This clustering trend was observed for both the Saint-Lazare (Figure 4a) and Rigaud aquifers (Figure 4b). PERMANOVA testing revealed that period (post-recharge and pre-recharged) significantly explained 25% of the variance (*p* = 0.004; Appendix A Appendix A) for the Saint-Lazare groundwater microbial community and 17% of the variance (*p* = 0.033; Appendix A Appendix A) for the community from the Rigaud aquifer.

Furthermore, we observed a drop in the calculated Shannon diversity index for bacterial communities sampled during the months associated with groundwater recharge. For the community from the Saint-Lazare aquifer, the diversity index decreased from 5.12 in February to 3 in March, while at the Rigaud aquifer it decreased from 5.39 in March to 2.23 in April (Appendix A Appendix A, Appendix A).

In terms of individual taxonomic groups, for the Saint-Lazare aquifer, bacterial communities were generally dominated by ASVs belonging to *Proteobacteria* and *Actinobacteriota* phyla (Appendix A Appendix A). Following groundwater recharge, ASVs associated with the phylum *Gemmatimonadota* represented a relatively important portion of all communities (average relative abundance of 6.6%) except for the one sampled in June. At the family level, communities sampled prior to groundwater recharge were generally dominated by ASVs associated with the families *Comamonadaceae*, *Moraxellaceae*, and *Beijerinckiaceae* (Figure 5a). Following snow melt, an important change in the composition of the bacterial community was noted. Notably, the March, April, July, and August communities were dominated by *Diplorickettsiaceae* (*Rickettsiaceae*) (relative abundances of 33.75, 34.86, 40.71 and 66.92%, respectively). *Ilumatobacteraceae* were also found in large abundance in the March and July samples (relative abundances of 15.93 and 20.55%, respectively).

Snow samples were generally dominated by ASVs associated with the families *Beijerinckiaceae* and *Acetobacteraceae* (mean relative abundance of 37.55 and 17.70%, respectively), except the community from the month of November, which was dominated by *Moraxellaceae* (relative abundance of 58.81%) (Figure 5b). Members of this family were also relatively abundant in groundwater communities sampled in September, October, and November (relative abundances of 13.32, 33.08, and 7.43%, respectively) (Figure 5a).

For the Rigaud aquifer, bacterial communities were also generally dominated by ASVs belonging to *Proteobacteria* and *Actinobacteriota* phyla (Appendix A Appendix A) and the composition of major phyla relatively more between months when compared to communities from the Saint-Lazare aquifer (Appendix A Appendix A). At the family level, communities sampled before groundwater recharge were mainly composed of the families *Acidiferrobacteraceae*, *Nitrospiraceae*, and *Oxalobacteraceae* (Figure 6a). Compared with the Saint-Lazare aquifer, bacterial communities from the Rigaud aquifer shared fewer taxa between the different months. Yet, similarly to Saint-Lazare, ASVs associated with the family *Diplorickettsiaceae* were largely represented in samples from April and August, but also June (relative abundances of 13.52, 31.66, and 46.48%, respectively). While the majority of ASVs belonging to the family *Diplorickettsiaceae* could not be classified at the genus level, 40.55 and 26.32% of *Diplorickettsiaceae* sampled in June and August, respectively, were associated with the genus *Aquicella*. June and August communities were also dominated by *Coxiellaceae* (28.02 and 48.38%, respectively), while ASVs associated with *Micrococcaceae* represented 41.26% of the July sample. Furthermore, *Beijerinckiaceae* dominated samples collected in March and April (relative abundances of 24.70 and 28.14%, respectively). Similar to Saint-Lazare, an important change in bacterial community composition was also noted in the months following the onset of snow melt. Communities from snow samples were also dominated by ASVs associated with the families *Beijerinckiaceae* and *Acetobacteraceae* (mean relative abundance of 17.38 and 13.56%, respectively) (Figure 6b).

### 3.3. Connectivity between Surface and Subsurface Environments

At the Saint-Lazare site, FEAST analyses suggest that previously sampled groundwater was the major contributor to current groundwater bacterial diversity (mean 71.54 ± 10.92%) (Figure 7a, Appendix A Appendix A). Soil contributed on average 2.50%, with the highest contributions occurring in October, November, December, and June (4.33, 2.26, 9.73, and 2.55%, respectively). Snow generally contributed less than 2% to the aquifer’s bacterial diversity, except in April, where the snow sampled in January contributed 18.93% to the groundwater bacterial diversity. The average proportion from unknown sources was 24.25 ± 8.29%.

For Rigaud’s aquifer, a larger proportion of diversity was attributed to unknown sources (mean 59.73 ± 22.17%) while previously sampled groundwater contributed on average 31.09 ± 19.57% (Figure 7b, Appendix A Appendix A). March and June had the lowest proportion of diversity derived from previous groundwater samples (2.66 and 2.59%, respectively). Generally, less than 1.25% of the composition of groundwater bacterial communities could be attributed to soil samples, except for July, where 10.35% of the diversity was derived from June’s soil sample. Contributions greater than 2% from river samples were noted in March, May, and July (5.36, 7.03, and 2.59%, respectively). Furthermore, 62.85% of the community sampled in March originated from snow samples.

We further investigated the migration of microbial communities from subsurface to surface environments with a second FEAST analysis where river samples were considered as sink. Due to a thick layer of ice covering the rivers, some samples could not be collected (the February sample at both sites and the March sample at the Rigaud site only). At both sites, most of the diversity was from unknown sources (Saint-Lazare: mean 62.78 ± 17.71%; Rigaud: mean 55.90 ± 20.98%). For the Saint-Lazare site (Figure 8a, Appendix A Appendix A), groundwater contributed on average 3.01% to river communities. Previously sampled bacterial communities from the river contributed on average 33.73% (±16.61%), with the largest contributions observed in January, March, and April (64.27, 44.88, and 53.26%, respectively). At the Rigaud site (Figure 8b, Appendix A Appendix A), 15.33% of the bacterial community sampled in October originated from the September groundwater, while the average contribution was 2.84% (±4.51%). The largest contribution of snow to the river occurred in January (10.05%). On average, 33.95% (±17.46%) of the bacterial river community originated from the previously sampled river.

## 4. Discussion

In groundwater systems, disturbances, often associated with recharge events, can greatly alter the composition and diversity of microbial communities [4,5,7,11]. These changes can be attributed to a modification of the abiotic conditions of the environment, but also to the introduction of surface-derived taxa to groundwater microbiomes. Considering the abundant and diverse microbial communities found in soils [34], they can be considered as a potential source of microbial taxa for subsurface environments [5,35]. In this study, we evaluated the impact of groundwater recharge on bacterial community composition and diversity in two aquifers with different hydrogeological settings, while also examining the potential connectivity between surface and subsurface environments, over the course of one year.

For both aquifers, we observed an important increase of groundwater level and a decrease of water temperature in March 2020. These changes could be attributed to seeping of melted snow into the subsurface. Following this recharge-related disturbance, changes in bacterial community composition and diversity were observed. Despite differences in hydrogeochemical conditions, these trends were noted for both sites.

For the Saint-Lazare site, these changes were mainly attributed to the large proportion of taxa belonging to the *Diplorickettsiaceae* family. Members from this family are obligate endosymbionts and parasites, which can infect a wide variety of eukaryotic species, such as protists, leeches, cnidarians, arthropods, and mammals [36]. While bacteria–protist relationships including members of the family *Rickettsiaceae* were reported in cooling towers [37] and acid mine drainage [38], bacteria–bacteria association was also noted between *Candidatus Patescibacteria* and *Rickettsiales* in groundwater from the Hainich Critical Exploratory Zone in Germany [39]. Interestingly, important relative abundances of ASV associated with *Diplorickettsiaceae* were also observed in groundwater sampled in July and August at the Saint-Lazare site, as well as in March, June, and August at the Rigaud site. More precisely, the majority of *Diplorickettsiaceae* identified in the June and August groundwater samples from the Rigaud site were associated with the genus *Aquicella,* while those from the Saint-Lazare site could not be classified at the genus level. Bacteria belonging to the genus *Aquicella* were found to proliferate in the amoeba *Hartmannella vermiformis* [40], which is widely distributed in the environment (water, soil, air, compost, and sediments) and mainly encountered in biofilms [41]. It is worth noting that the majority of microorganisms in aquifers are not present as free-floating or planktonic communities but are rather present in micro-colonies or biofilms attached to rock surface or sediment particles [3,42]. During recharge events or during periods of important aquifer–river connections, increases in groundwater flow and level could cause these colonies to detach and enter the water column [12]. This could explain the increased abundance of *Aquicella*. Although not included in this study, it is also possible that other Eukaryote hosts of unclassified *Diplorickettsiaceae* benefited from certain changes in the physicochemical parameters of the groundwater during recharge and summer months, which could explain their increased abundance.

In a previous study by Yan et al. [5], the authors generally observed a significant increase in diversity following groundwater recharge. The authors explained this phenomenon by suggesting the dissemination of microorganisms from the permanently saturated zone and the soil into the aquifer. In accordance, our results from microbial source tracking at the Rigaud site suggest that the majority of the bacterial community sampled in March originated from the snow (62.85%) and the river (5.36%), while only a small proportion (0.23%) originated from the soil. These results could be explained by the important relative abondance of ASVs belonging to the families *Beijerinckiaceae*, *Micromonosporaceae*, *Pseudonocardiaceae*, *Sphingomonadaceae*, and *Acetobacteraceae*, which were also identified in both the March groundwater sample and January snow sample (Figure 6b).

Unfortunately, as mentioned previously, given the thick layers of ice and snow we were not able to obtain samples from both soil and river habitats during the month of March. Therefore, we cannot confirm if the bacterial communities from these habitats were similar in terms of composition to the ones from the January snow sample and March groundwater sample. Thus, we cannot determine if surface-derived microorganisms reached the aquifer through the soil or through the groundwater–river connection. Yet, we can assume that the three concurring events (rise in groundwater level, drop in water temperature, and transfer of microorganisms from snow to aquifer) are all caused by the melting of the snow from March to April. Using microbial source tracking, this connection between snow and aquifer is potentially less evident for the Saint-Lazare site, considering that members of the family *Beijerinckiaceae*, which account for almost 50 % of snow bacterial communities (Figure 5b), were also consistently present in groundwater samples. Nonetheless, bacterial community from this aquifer also exhibited strong changes in terms of composition during both months associated with snow melt.

Given the less permeable nature of the clay soil from the Rigaud site, we expected the aquifer to be relatively isolated from surface influences, and thus more stable across time. Surprisingly, we observed greater variation in both environmental parameters (Figure 2b,d) and bacterial community composition (Figure 5a) of the groundwater when compared with the Saint-Lazare aquifer. This could suggest a potential connectivity between subsurface and surface environments in an area located outside our sampling site. This hypothesis could also explain the large proportion of diversity originating from unknown sources. As mentioned earlier, it is also possible that increase in groundwater flow and level could cause sessile colonies to detach and enter the water column. Therefore, sessile microorganisms from the subsurface could act as a microbial seedbank for planktonic communities during periods of groundwater recharge and explain the large proportion of diversity from unknown source [5].

Furthermore, our results from microbial source tracking also suggest a minor contribution of subsurface environments to river communities through groundwater discharge. These results are consistent with a previous study by Villeneuve et al. [30]. Our results also confirmed that most of the bacterial diversity for rivers was attributed to unknown sources, which can be explained by rapid changes in community composition as a result of water transit [43].

## 5. Conclusions

In this study, we set out to evaluate the temporal variations in microbial community composition and the migration of microorganisms from surface to subsurface and back to the surface in two aquifers with different hydrogeochemical settings. Overall, despite the differences in soil permeability, we observed the same patterns for both aquifers beginning in March: a rise in water level, a drop in water temperature, an important contribution of snow to groundwater bacterial diversity, and a drastic change in the composition of bacterial communities. We suggest that these changes were induced by the onset of snow melt. Interestingly, the Rigaud aquifer, which we assumed to be more isolated from the surface owing to the less permeable nature of its clay soil, showed greater variations in both microbiome and abiotic conditions. Yet, we could not explain these fluctuations using microbial source tracking with the adjacent river and top layer of the soil as potential sources. Furthermore, despite differences in bacterial community composition and abiotic conditions between both aquifers, ASVs associated with the endosymbiotic and parasitic family *Diplorickettsiaceae* had an important relative abundance in samples collected after March. However, in order to evaluate the presence of potential hosts of *Diplorickettsiaceae* and assess the lingering effect of the punctual disturbance associated with snow melt, we believe a longer temporal study including both Procaryotes and Eukaryotes would be required.

## Figures and Tables

**Figure 1 microorganisms-11-01526-f001:**
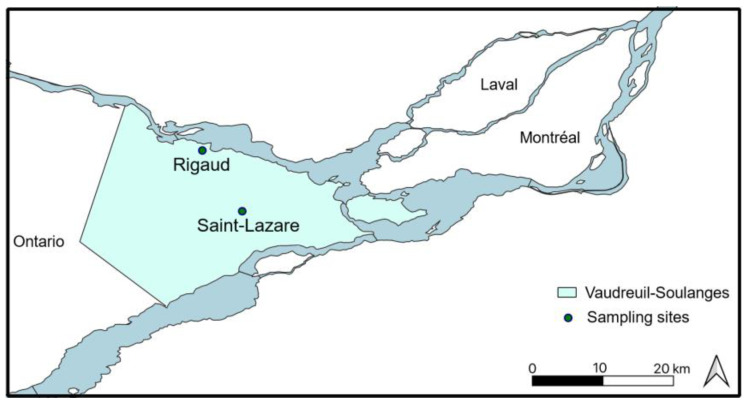
Location of the two aquifer sampling sites. Sampling sites are marked with green circle and the Regional County Municipality of Vaudreuil–Soulanges, Quebec (Canada) is highlighted in blue.

**Figure 2 microorganisms-11-01526-f002:**
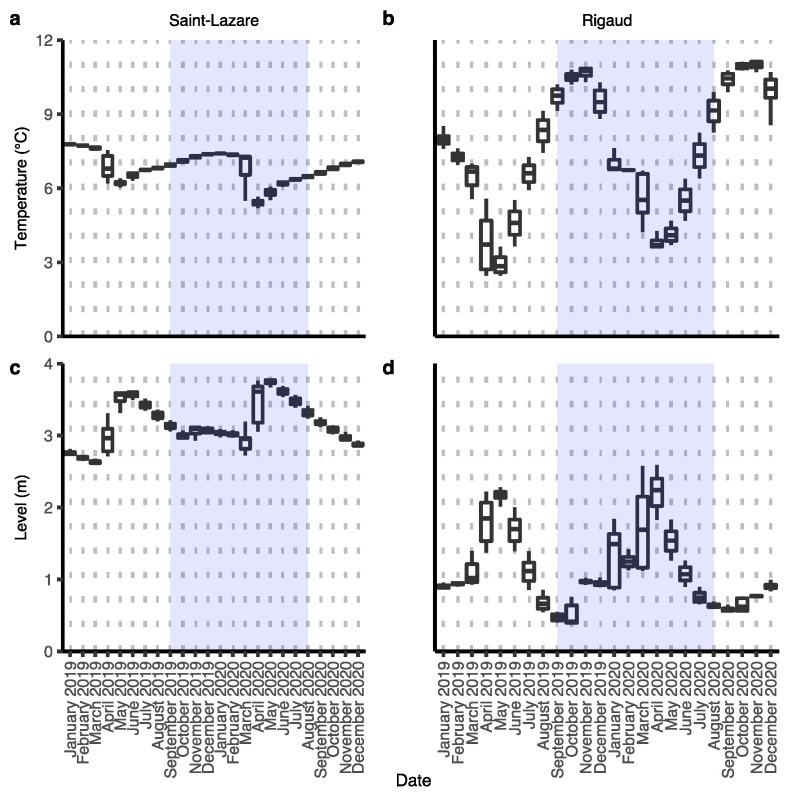
Variations in groundwater temperature and level of both aquifers. Data were collected by a probe installed semi-permanently in each well. Hourly measures were averaged for each date. Level indicates the height in meter of the water above the probe. Sampling period from this study is highlighted in blue. Temperature: Saint-Lazare (**a**) and Rigaud (**b**); groundwater level: Saint-Lazare (**c**) and Rigaud (**d**).

**Figure 3 microorganisms-11-01526-f003:**
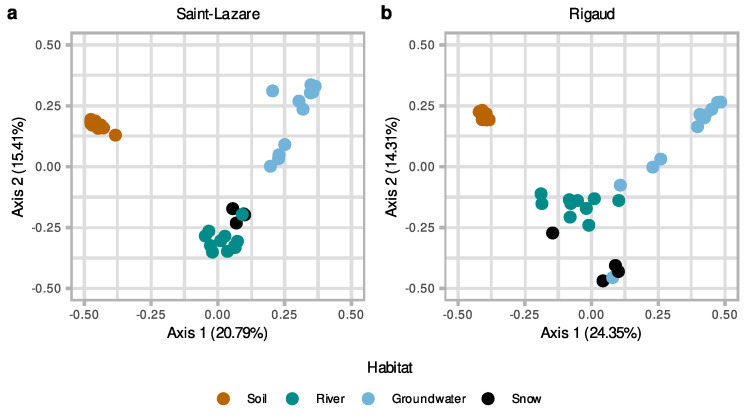
Differences in microbiome composition as a function of habitat for both sites. Principal coordinate analysis (PCoA) ordination of bacterial community composition generated with a Bray–Curtis dissimilarity matrix based on the rarefied and Hellinger-transformed relative abundance matrix for all samples from the Saint-Lazare site (**a**) and the Rigaud site (**b**).

**Figure 4 microorganisms-11-01526-f004:**
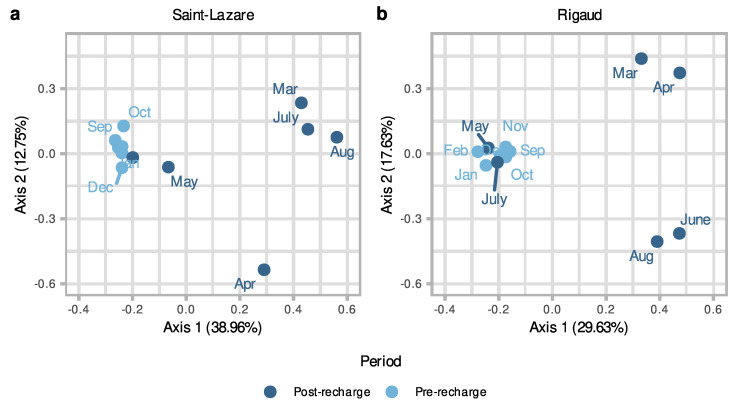
Differences in groundwater microbiome composition as a function of period for both sites. Principal coordinates analysis (PCoA) ordination of bacterial community composition generated with a Bray–Curtis dissimilarity matrix based on the rarefied and Hellinger-transformed relative abundance matrix for groundwater samples from the Saint-Lazare site (**a**) and the Rigaud site (**b**).

**Figure 5 microorganisms-11-01526-f005:**
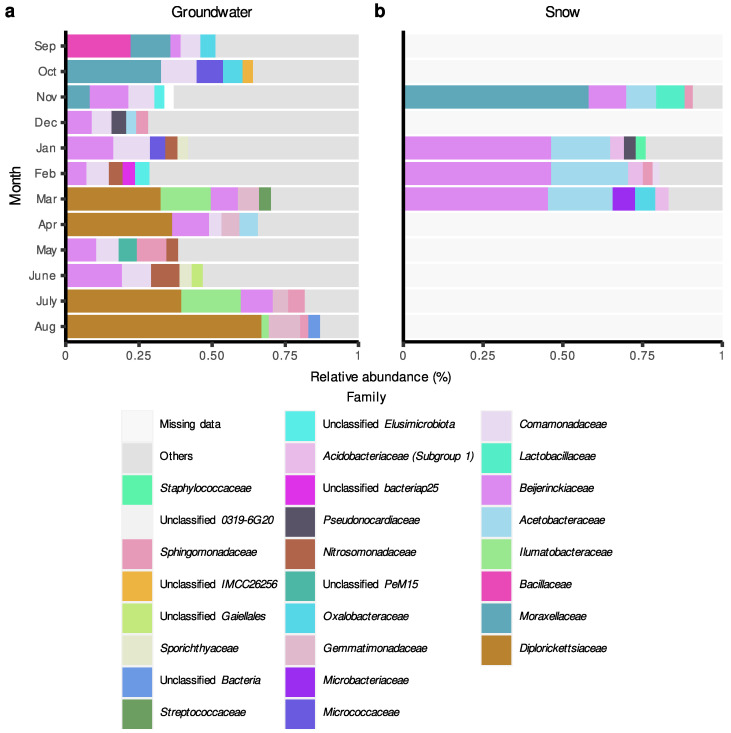
Relative abundance (%) of amplicon sequence variant (ASV) at the family level in groundwater and snow samples collected monthly at the Saint-Lazare site. For each month, only the five most abundant families are colored. (**a**) Groundwater samples. (**b**) Snow samples.

**Figure 6 microorganisms-11-01526-f006:**
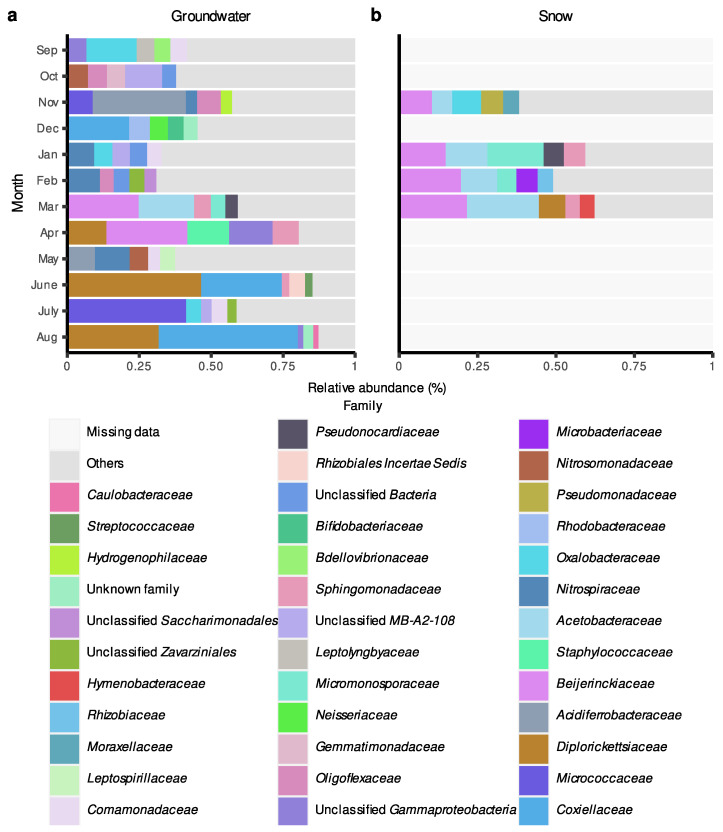
Relative abundance (%) of amplicon sequence variant (ASV) at the family level in groundwater and snow samples collected monthly at the Rigaud site. For each month, only the five most abundant families are colored. (**a**) Groundwater samples. (**b**) Snow samples.

**Figure 7 microorganisms-11-01526-f007:**
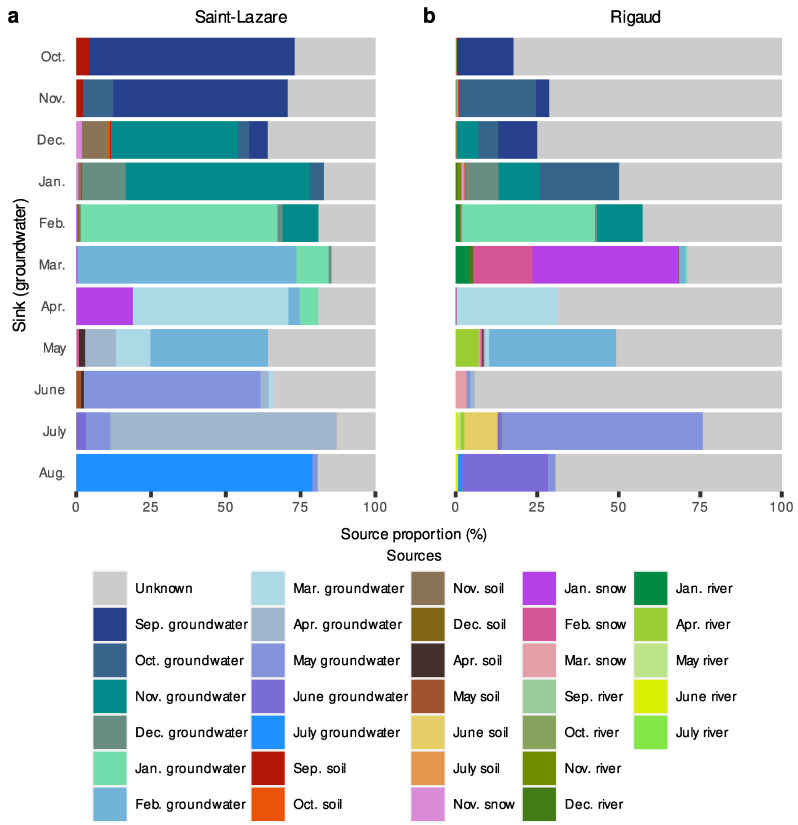
FEAST estimations of source contribution to groundwater microbiomes. Stacked-bar plots depicting the estimated contribution in percentage of each source (previously sampled groundwater, soil, snow, and river samples) to the formation of groundwater communities for each month (October 2019 to August 2020) at the Saint-Lazare (**a**) and Rigaud (**b**) sites. For each sink, we only considered the contribution from samples collected in the previous three months.

**Figure 8 microorganisms-11-01526-f008:**
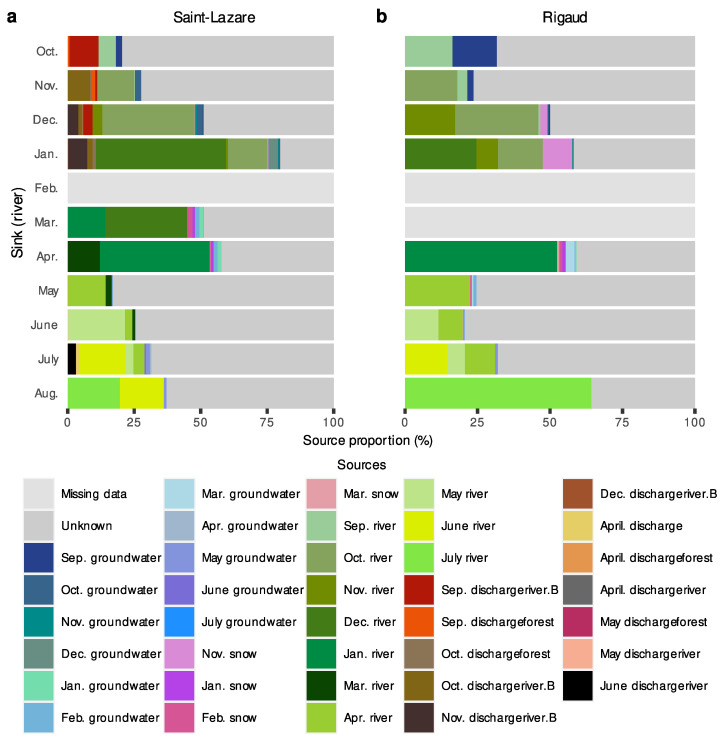
FEAST estimations of source contribution to river microbiomes. Stacked-bar plots depicting the estimated contribution in percentage of each source (previously sampled river, groundwater, snow, and samples) to the formation of river bacterial communities for each month (October 2019 to August 2020) at the Saint-Lazare (**a**) and Rigaud (**b**) sites. For each sink, we only considered the contribution from samples collected in the previous three months.

## Data Availability

All R scripts and extra metadata used in this study are available on GitHub at https://github.com/karinevilleneuve/Groundwater_surface_connectivity (accessed on 20 March 2023). The obtained sequences were deposited on the National Center for Biotechnology Information platform (NCBI) under the BioProject ID: PRJNA912985.

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
