# Peer review of "Effect of Snowmelt on Groundwater Bacterial Community Composition and Potential Role of Surface Environments as Microbial Seed Bank in Two Distinct Aquifers from the Region of Quebec, Canada"

_microorganisms, 2023, doi:10.3390/microorganisms11061526_

Round 1

Reviewer 1 Report

This paper by Villeneuve et al. deals with aquifers' microbiomes in a region of Québec, Canada, and is aimed to investigate, among other geophysically relevant issues, the migration of microorganisms from surface to subsurface and back. They found that the  bacterial community composition of the studied aquifers significantly differed between samples collected prior and after groundwater recharge by melting snow. The method used for the microbiome analysis was 16S rRNA gene sequencing focusing on bacterial gene sequences. In the Methods section, the analysis of the rRNA sequences was followed by taxonomic assignment by mapping sequences against the trained SILVA SSU database. No further detail is given/ However, Figs. 5 and 6 report bacterial taxa at the family level – ONLY. There is no taxonomic breakdown of the taxa along major phyla, for instance Proteobacteria vs. Actinobacteria, which dominate soil environments. These figures are poorly drawn and confusing, given the similar coloring of various taxa/families. 

The major findings of microbiological relevance reported are that in certain months, after aquifer recharge, bacterial communities are dominated by Diplorickettsiaceae. Diplorickettisaceae belong to the class of Gammaproteobacteria and order of Legionellales, in the great majority endocellulr parasites. Indeed, Diplorickettisaceae taxa are abundant in the gut microbiome of cocroaches:

Tinker, K. A., & Ottesen, E. A. (2021). Differences in gut microbiome composition between sympatric wild and allopatric laboratory populations of omnivorous cockroaches. Frontiers in Microbiology, 12, 703785.

So, the abundance of these endocellular parasites must be related to a large increase in the population of eukaryotes (please correct the wrong spelling of this term at lane 430) polluting the aquifers. This problem is recognized in the Conclusion of the paper, lanes 423-431. However, it is not sufficient to recognize the problem and defer it to subsequent work. The Authors need to dig into their data at least at the genus level, so as to define which taxa of parasitic Diplorickettsiaceae are most abundant, and then from their taxonomic profile deduce from which eukaryotic source they may originate. In this way they will add microbiological value to their work, otherwise essentially relevant to geochemical issues.

The paper is concisely written in good style, overall

Author Response

We would like to thank the reviewer for taking the time to review our study and for her/his comments.

Point 1: There is no taxonomic breakdown of the taxa along major phyla, for instance Proteobacteria vs. Actinobacteria, which dominate soil environments. These figures are poorly drawn and confusing, given the similar coloring of various taxa/families. 

Response 1: We have included in the supplementary material histograms representing the relative abundances of the most abundant phyla (Figure S3 and S4). Description was also included in the main text at line 269-273 and 291-294. We would argue here that these types of figures are commonly used in study of microbial communities and given the important number of taxa it is difficult to avoid similar colors.

Point 2: Please correct the wrong spelling of this term (Eukaryotes) at lane 430.

Response 2: Spelling was corrected at line 462 and 399.

Point 3: The Authors need to dig into their data at least at the genus level, so as to define which taxa of parasitic Diplorickettsiaceae are most abundant, and then from their taxonomic profile deduce from which eukaryotic source they may originate.

Response 3: We found that most ASVs associated to family Diplorickettsiaceae belonged to genus Aquicella or were unclassified as this taxonomic level. Further details were provided in the main text at line 300-302 of the Result section and 387-398 of the Discussion section.   

Reviewer 2 Report

Manuscript submitted by Villeneuve et al. includes a significant information on microbial ecology. The drafting of manuscript is good along with an effective discussion of results. I have some minor comments to improve the manuscript:

1. Sampling design needs to be elaborated and explained clearly. For example, how many samples exactly used for the microbiome analysis from various sources.

2. In NGS, author's need to provide the raw reads obtained and filtered quality reads used in the analysis (total no. of reads), in the main text of methods or results.

3. Figure legend 2. Line no. 205: It is mentioned that “highlighted in blue””: I didn’t understand how it is recognizable. Please replace this annotation with a sign not with color. As I cant see any color in the figure, except black.

4. Name of taxonomic level such as family should be italicized. Please make corrections in Figure 6. Also check throughout the text.

5. Figure quality can be improved by appropriately placing the legend and by using the identical fonts in the figure(s).

6. Introduction can be improved with some relevant recent (2022-2023) references.

Author Response

We would like to thank the reviewer for taking the time to review our study and for her/his comments.

Point 1. Sampling design needs to be elaborated and explained clearly. For example, how many samples exactly used for the microbiome analysis from various sources.

Response 1: The total number of samples as well as the amount from different sources was added at line 105-106.

Point 2. In NGS, author's need to provide the raw reads obtained and filtered quality reads used in the analysis (total no. of reads), in the main text of methods or results.

Response 2: Total number of raw and filtered reads was added at line 166-167.

Point 3. Figure legend 2. Line no. 205: It is mentioned that “highlighted in blue””: I didn’t understand how it is recognizable. Please replace this annotation with a sign not with color. As I cant see any color in the figure, except black.

Response 3: We believe conversion between mac and windows caused this absence of color. We corrected this issue by using extension SVG rather than PDF for all figures.  

Point 4. Name of taxonomic level such as family should be italicized. Please make corrections in Figure 6. Also check throughout the text.

Response 4: Correction were made for figures 5 and 6 and we also reviewed the main text.

Point 5. Figure quality can be improved by appropriately placing the legend and by using the identical fonts in the figure(s).

Response 5: Font size was adjusted for every figures.

Point 6. Introduction can be improved with some relevant recent (2022-2023) references.

Response 6: We would argue here that most relevant recent articles were included in the introduction.

Reviewer 3 Report

The article presents an interesting analysis of microbial diversity in two aquifers. The results are attractive, as they discuss the effect of snowmelt and the contribution of bacterial microbiota in these environments.

In general:

The authors should present the same sections of the methodology in the results. Particularly, the authors do not start with physicochemical analyses, but rather with the analysis of temperature and diversity.

In addition, they do not discuss anything regarding the chemical characterization of the water described in lines 123-132. The authors need to discuss.

Families are incorrectly written in italics throughout the entire manuscript. Italics are only used for genera and species.

Revise the endings of the families, as some correspond to orders, as well as phyla.

Remove italics from the families on the figures.

The proportions of Gracilicutes and Firmicutes found can be discussed.

Specific comments

In the introduction, it is worth mentioning terms such as "psychrophilic", "psychrotrophic" and "mesophilic", whose presence in these types of environments is the feasible diversity to be found.

Regarding sampling, they mention that they collected 4 liters for microbiological analysis and refer to serial filtrations. How many filters were used for that amount of sample?

In line 139, which is methodology, the authors refer not having found microorganisms of the Archaea Domain, this is a result. Mention in the corresponding section.

line 232 is the first time they use SL for the site, it is not necessary since no more abbreviations are used in the manuscript.

lines 267 to 284 separate paragraph to make it easier to read.

Revise reference numbers with respect to those cited in the manuscript.

In the discussion section although the authors did not find Archaea, mention what they attribute methodologically  and the prospection to isolate prokaryotes of this type.

Author Response

We would like to thank Reviewer’s for taking the necessary effort and time required to review our manuscript. We sincerely appreciate all comments and suggestions, which help improve the quality of the manuscript. We carefully studied the comments and have made corrections which we hope will be met with approval.

Point 1: The authors should present the same sections of the methodology in the results. Particularly, the authors do not start with physicochemical analyses, but rather with the analysis of temperature and diversity. In addition, they do not discuss anything regarding the chemical characterization of the water described in lines 123-132. The authors need to discuss.

Response 1: A short paragraph discussing the physicochemical analyses results was added before the results of the diversity analyses (line 235-237: “Other physicochemical qualities of groundwater from both aquifers were relatively stable across the sampling period, with the exception of a noticeable increase in dissolved inorganic carbon during the month of March for both sites (Table S2).”). Furthermore, the title of the section was changed from “Groundwater level and temperature fluctuation” to “Fluctuation in groundwater level, temperature and physicochemical qualities”. Methodology and results are therefore presented in the same order.

Point 2: Families are incorrectly written in italics throughout the entire manuscript. Italics are only used for genera and species. Revise the endings of the families, as some correspond to orders, as well as phyla. Remove italics from the families on the figures.

Response 2: As MDPI does not specify the preferred nomenclature for bacterial taxa, we followed the guidelines from the American Society for Microbiology (ASM) (https://journals.asm.org/nomenclature) as well as the instruction from other Reviewers to print in italics the family names in the text and figures.

Point 3: The proportions of Gracilicutes and Firmicutes found can be discussed.

Response 3: We would argue here that while the ratio of Firmicutes to Bacteroidetes ratio is of significant relevance in studies of the human gut, the relevance of such ratio in studies focusing on microbial ecology of groundwater systems has not been demonstrated. We therefore do not believe discussing such ratio would benefit the manuscript.

Point 4: In the introduction, it is worth mentioning terms such as "psychrophilic", "psychrotrophic" and "mesophilic", whose presence in these types of environments is the feasible diversity to be found.

Response 4: We acknowledge the idea of mentioning these terms, but we deem such addition to be out of scope of our research.  

Point 5: Regarding sampling, they mention that they collected 4 liters for microbiological analysis and refer to serial filtrations. How many filters were used for that amount of sample?

Response 5: Precision on the amount of filter was added at lines 113-114.

Point 6: In line 139, which is methodology, the authors refer not having found microorganisms of the Archaea Domain, this is a result. Mention in the corresponding section.

Response 6: We would argue here that we do not consider such information as a result but rather as an explanation to why our research only focused on microorganisms from the Bacterial domain.

Point 7: line 232 is the first time they use SL for the site, it is not necessary since no more abbreviations are used in the manuscript.

Response 7: The SL abbreviation was removed and replaced by Saint-Lazare (see line 243).

Point 8: lines 267 to 284 separate paragraph to make it easier to read.

Response 8: Paragraph was separated at line 291.

Point 9: Revise reference numbers with respect to those cited in the manuscript.

Response 9: Reference numbers were revised and all reference match those cited in the manuscript.

Point 10: In the discussion section although the authors did not find Archaea, mention what they attribute methodologically and the prospection to isolate prokaryotes of this type.

Response 10: While we do believe that Archaea are important members of microbial communities and should be included, when possible, we would argue there that such addition to the discussion section would unnecessarily burden the discussion.